# miRTil: An Extensive Repository for Nile Tilapia microRNA Next Generation Sequencing Data

**DOI:** 10.3390/cells9081752

**Published:** 2020-07-22

**Authors:** Luiz Augusto Bovolenta, Danillo Pinhal, Marcio Luis Acencio, Arthur Casulli de Oliveira, Simon Moxon, Cesar Martins, Ney Lemke

**Affiliations:** 1Department of Biophysics and Pharmacology, Institute of Biosciences of Botucatu, São Paulo State University—UNESP, Botucatu SP CEP:18618-689, Brazil; ney.lemke@unesp.br; 2Department of Chemical and Biological Sciences, Institute of Biosciences of Botucatu, São Paulo State University—UNESP, Botucatu SP CEP:18618-689, Brazil; danillo.pinhal@unesp.br (D.P.); arthur.c.oliveira@unesp.br (A.C.d.O.); 3Luxembourg Centre for Systems Biomedicine, Université du Luxembourg, Campus Belval, 7, Avenue das Hauts-Fourneaux, L-4362 Esch-sur-Alzette, Luxembourg; marcio.acencio@uni.lu; 4School of Biological Sciences, University of East Anglia (UEA), Norwich Research Park, Norwich NR4 7TJ, UK; s.moxon@uea.ac.uk; 5Department of Structural and Functional Biology, Institute of Biosciences of Botucatu, São Paulo State University—UNESP, Botucatu SP CEP:18618-689, Brazil; cesar.martins@unesp.br

**Keywords:** database, microRNA, post-transcriptional regulation, Nile tilapia, microRNA target prediction, microRNA expression profile

## Abstract

Nile tilapia is the third most cultivated fish worldwide and a novel model species for evolutionary studies. Aiming to improve productivity and contribute to the selection of traits of economic impact, biotechnological approaches have been intensively applied to species enhancement. In this sense, recent studies have focused on the multiple roles played by microRNAs (miRNAs) in the post-transcriptional regulation of protein-coding genes involved in the emergence of phenotypes with relevance for aquaculture. However, there is still a growing demand for a reference resource dedicated to integrating Nile Tilapia miRNA information, obtained from both experimental and in silico approaches, and facilitating the analysis and interpretation of RNA sequencing data. Here, we present an open repository dedicated to Nile Tilapia miRNAs: the “miRTil database”. The database stores data on 734 mature miRNAs identified in 11 distinct tissues and five key developmental stages. The database provides detailed information about miRNA structure, genomic context, predicted targets, expression profiles, and relative 5p/3p arm usage. Additionally, miRTil also includes a comprehensive pre-computed miRNA-target interaction network containing 4936 targets and 19,580 interactions.

## 1. Introduction

Nile tilapia (*Oreochromis niloticu*s) is an economically important Teleostei species belonging to the African family of Cichlidae. According to the Food and Agriculture Organization of the United Nations [1], Nile tilapia is the third most farmed freshwater fish in the world and is especially relevant for developing countries [2,3]. Due to its economic potential, the scientific community is interested in revealing the molecular mechanisms underlying economically interesting genotypes. Some traits include sexual differentiation, muscle growth, lipogenesis, and disease resistance [4,5,6,7]. Data from the plant science community have shown that economically interesting traits, such as disease resistance, biomass production, plant development, and environmental biotic or abiotic stress, can be altered by manipulation of microRNAs (miRNAs) [8,9,10]. These endogenous small RNA molecules of approximately 22 nucleotides in length are post-transcriptional regulators of gene expression acting by pairing completely or partially to messenger RNAs (mRNAs), especially in the 3’-untranslated regions (3’ UTR) [11,12,13].

miRNAs have been associated with economically relevant Nile tilapia traits, such as sexual differentiation [14,15,16,17], muscle growth, and disease resistance [18,19,20]. However, most of Nile tilapia-specific miRNA data is available only in textual form in biomedical literature [15,16,17,18,21,22,23,24]; therefore, scientists must use non-automated workflows to establish relationships between miRNA expression profiles and economically relevant traits. To facilitate the implementation of automated workflows and other large scale analysis, it is imperative to provide Nile tilapia-specific miRNA data in an open-access and user-friendly database. To the best of our knowledge, currently, there is no specialized Nile tilapia miRNAs database.

Inspired by other species-specific miRNA databases, such as Foxtail millet microRNA database (FmMiRNADb) [25], The Apple Gene Function & Gene Family DataBase (AppleGFDB) [26], and Wheat miRNA Database (WMP) [27], we developed an open-access database designed to host Nile Tilapia miRNA data, the "miRTil database". The miRTil database provides, for each miRNA, its basic structure, genomic coordinates and context characterization (intronic, exonic, or intergenic), predicted targets and expression profiles in all tissues, and the relative abundance of mature miR-5p and miR-3p. In addition, miRTil also offers a predicted miRNA-target interaction list, containing gene targets (following Oliveira et al. [28]). In this paper, we describe the miRTil database and present a case study. The miRTil database is freely available at https://www.lbbc.ibb.unesp.br/mirtil.

## 2. Results

### 2.1. Database Structure

miRTil was developed using a relational database structure. The database was designed to allow storage and retrieval of Nile tilapia miRNAs data including specimen details, miRNA molecule identification, annotation (using oreNil2 genome version), expression profiles, and putative targets. The expression profile dataset considers 16 samples from several adult tissues and developmental stages with detailed expression values for 5p and 3p mature miRNAs. miRTil also contains tissue and developmental expression evidence on predicted gene targets and conserved interactions of miRNA-mRNA interactions in humans and zebrafish. Additionally, the database provides information about an miRNA’s precursor and mature molecules, including sequence size, secondary structure, CG content, and minimum free energy (MFE) for all pre-miRNA loci. For future miRTil versions, the database structure was also designed to store information about miRNA flanking genes, as well as their transcriptional regulatory roles (Figure 1).

### 2.2. Database Content

miRTil contains 734 mature miRNAs identified in samples, including adult tissues and developmental stages. miRNA containing low read counts (an expression level less than five raw reads in more than 75% of the samples) are classified as “low confidence” miRNAs. On the contrary, abundant miRNAs (an expression level larger than five raw reads in at least 25% of the samples) are classified as “high confidence”. Of these 734 mature miRNAs, 567 are high confidence miRNAs and 167 are low confidence miRNAs. The high confidence miRNA set contains 470 conserved mature miRNAs (243 miRNA-5p and 227 miRNA-3p) and 97 novel mature miRNAs (60 miRNA-5p and 37 miRNA-3p). The low confidence miRNA set contains 70 conserved and 97 novel miRNAs.

The predicted targets comprise 4936 mRNAs and 19,580 interactions with *O. niloticus* genes. Of these 19,580 interactions, 15,902 correspond to unique mature miRNAs ignoring the numbers generated by the presence of paralogous miRNAs copies and 14,683 interactions considering just one microRNA-recognition element (MRE) to each miRNA in the targets.

There are approximately 5000 miRNA family interactions with Target Scan Context+ at exactly −0.250. Briefly, the Context+ Score is a score created amongst the union of several characteristics in the miRNA interactions [29]. It evaluates the binding probability and efficiency of a predicted interaction between the miRNA and the target, since this score may be rankable, and the lowest values symbolize the highest probability and efficiency of binding. Our results show a data concentration in a range of values considered ideal by other groups [30,31] (Appendix A).

The lowest score was −0.618 for two interactions involving miR-184-3p and miR-184-3p-2 with one novel Nile tilapia transcript (ENSONIT00000021503) predicted to code for a protein similar to glycerol-3-phosphate dehydrogenase (Figure 2), an enzyme that plays biological roles in carbohydrate metabolism and lipid metabolism [32,33]. In general, the predicted miRNA-mRNA interaction network shows a typical power-law degree distribution, i.e., a few miRNAs control many targets and many miRNAs control a few targets. For example, in the Nile tilapia network, one miRNA is predicted to regulate 1439 mRNAs, while 11 miRNAs might interact with less than 24 mRNAs (Appendix A). This result is following previous studies showing that a few miRNAs may control many genes [34].

Finally, we show the localization of 303 miRNA genes within Nile tilapia linkage groups (LGs) that represent the 22 putative chromosomes of Nile tilapia. Moreover, we included an additional 19 putative miRNA genes obtained from Huang et al. and YAN et al. [18,22]. Sixty-five miRNA genes from all sources were mapped to genome scaffolds not yet anchored to any of the LGs; as a consequence, they lack information about their genomic coordinates. We identified 41 putative miRNAs clusters (tandem miRNAs) and 51 miRNAs mapped to a single locus. In addition, we found that 179 miRNAs derive from intergenic, and 142 from intronic, regions. We also found that 47 miRNAs are derived from exonic regions, 40 of which are within annotated genes distributed in 15 distinct LGs. The remaining 7 miRNAs are within predicted genes not yet allocated into LGs. Among 47 exonic miRNAs, 13 are putative novel miRNA genes.

### 2.3. Data Access and Interface Functionalities

miRTil is accessed using a web interface (https://www.lbbc.ibb.unesp.br/mirtil) that allows a user-friendly and intuitive view of miRNA data. There are two mechanisms for information retrieval: a quick search (Figure 3) and an advanced user interface (Figure 4).

The quick search mechanism allows fast access to miRNA data using just the identification number (ID) or the complete/partial miRNA symbol name. The advanced user search mechanism offers three search options: (1) miRNA name or identification number, (2) genomic location, or (3) the analyzed sample. In option 1, it is necessary to use the identification number (ID) or the complete/partial miRNA name. In option 2, users should provide the LG and the genomic coordinates. Finally, in option 3, miRNAs of interest are retrieved by informing the sample type, gender, and developmental stages. In addition, all detailed information about the retrieved miRNAs, e.g., protein-coding gene host, can be accessed in miRTil at the miRNA details page, described in the section “Case Study”.

miRTil datasets can be downloaded in different formats: miRNAs expression level values can be downloaded in Excel format (.xls), miRNA and mRNA target information can be downloaded in text format (.txt) or comma-separated values format (.csv), and mature and precursor miRNA sequences can be downloaded in FASTA format.

miRTil also allows genomic visualization of miRNA on LGs. This information is available in an interactive map that allows us to highlight paralogs and cluster miRNAs genes and detailed information about them (Figure 5). In the case of miRNA paralog genes, they are highlighted showing their genomic coordinates in neighbor LGs. Another feature is associated with miRNAs in tandem sequence (clusters), we can access detailed information about all miRNAs present in the tandem sequence by clicking on the root point (line) on the genomic map.

miRTil update policy will follow a year-to-year update rule, and old releases will be maintained for download. The links for database flat-files and modification description files are presented on the front page (Figure 3).

### 2.4. Case Studies

Here, we will show ways to retrieve information using miRTil. We choose “oni-miR-1-3p” as a miRNA candidate. The first step is to choose the advanced search mechanism among the three options for miRNA data retrieval: (i) ID or miRNA name symbol; (ii) genomic location; or (iii) sample used for miRNA identification processes (Figure 4).

#### 2.4.1. ID or miRNA Name Symbol Search Mechanism

This search mechanism is indicated for users interested in a specific or few miRNAs. In this case study, we used the miRNA name symbol as an option. We used “oni-miR-1-3p” as the miRNA symbol name search. Optionally, the user can use partial a name, e.g., “oni-miR-1”, but it will return all miRNAs with “oni-miR-1” present in the symbol, such as “oni-miR-133” or “oni-miR-1-5p”.

The first result page is an intermediate page showing all alternative miRNAs that contain summarized search results (Figure 6). This page shows simple information about miRNA, such as IDs, miRNA symbol name, mature sequences, and genomic location (LG or scaffold number, start, end, and DNA strand). All information can be expanded to show advanced content with details of the specific miRNA. All miRNAs displayed are linkable to the miRNAs details page.

#### 2.4.2. Genomic Location Search Mechanism

If the user is interested in genomic features associated with a miRNA, such as the determination of miRNA members in a miRNA tandem sequence (cluster), detection of genes flanking a miRNA, or host genes for intragenic miRNAs, we recommend using the genomic location search mechanism. Here, the user can retrieve miRNA information using the LG and the coordinates of the start and end nucleotides to determine a genomic window. All miRNAs present in this window will be returned independent of strand sense.

We can select any genomic window region of interest; in this case, we selected a window from “LG18” in a range of 14,480,000–14,490,000 bps (Appendix A). As an outcome, we retrieved 2 miRNA genes: oni-mir-133a-1 and oni-mir-1-1 (Appendix A), which represent a putative miRNA cluster already described in other species (e.g., *Danio rerio*). Furthermore, a paralogous copy of mir-1/133 could be accessed using the genomic frame region from “LG17” in a range of 5,216,200–5,233,500 bps.

#### 2.4.3. Sample Search Mechanism

Another user case is searching for specific samples to identify where a given miRNA is expressed or for users trying to detect biological pathways occurring on a given sample. For both cases, we strongly recommend the use of a sample search mechanism. The users can select all miRNAs expressed in the samples available in miRTil, as well as the gender and the developmental stage of Nile tilapia used for small RNA extraction.

Initially, we might choose miRNAs among 16 expression sample profiles (as described in Pinhal and Bovolenta et al. [24]) by selecting the tissue or developmental stage, gender (male, female, or sex mixed, described as “pool”), and age of individuals used for the miRNA-seq (Appendix A). In this case study, we selected “white muscle” as the sample, “male” as gender, and “Adult” (260 days) as individual age; in this example, we retrieved a total of 532 mature miRNAs expressed in male white skeletal muscle, partially represented in Appendix A.

All returned miRNA results can be expanded to show miRNA details in the advanced content page (Figure 7). This page displays information about (i) the mature miRNA, miRNA precursor and loop sizes (number of nucleotides); (ii) mature and precursor sequences; (iii) secondary structure and MFE; (iv) miRNA detection method; (v) mature region type; genomic information; and (vi) GC content. On this page, two expandable areas allow the visualization of regulatory information and details about the experimental identification method (sample information) (Figure 7).

In the experimental section, users can find information about the sample in which oni-miR-1-3p was identified, such as species, gender, developmental stage, sample type, the description of miRNA identification methodology, and miRNA expression levels.

In the regulatory section, information associated with predicted or experimentally verified miRNA-mRNAs interactions are displayed. This includes gene identifier and transcript accession number (Ensembl [36]), tissues with significant target gene expression levels, and TargetScan prediction information (prediction software version, binding site types, context+ score values, and MRE region). In the case of multiple samples or detection of multiple predicted targets, the information block is replicated by the number of retrieved items (Appendix A).

## 3. Discussion

miRTil was created as a complementary data source to miRBase to integrate information about Nile tilapia miRNAs [35] and to help scientists interested in understanding miRNA regulatory mechanisms. The database specializes in presenting miRNA information obtained from RNA-seq techniques and results from bioinformatics analysis, such as target prediction and gene expression profiling.

The database offers information about conserved miRNA-mRNA interactions in three different organisms: zebrafish, humans, and Nile tilapia, which can be explored under an evolutionary perspective of comparative genomics. Furthermore, these data are useful if used in conjunction with protein-coding gene expression data to improve the precision of interactions retrieved after prediction. Although the target prediction tools focus on the prediction specificity (i.e., TargetScan [28,37]), the occurrence of false-positive predictions is a common drawback [38]. The comparison of the protein-coding gene to miRNA expression levels in the same sample, as well as the conservation of miRNA-MRE interactions among different species, increases the probability of success in the experimental validation step. On the other hand, the conservation-based prediction strategy can exclude species-specific regulatory interaction due to the evolutionary distances between species [39]. miRTil also provides expression profiles for several tissues and developmental stages, which allows the identification of regulatory functions and the detection of expression profile correlations between samples, providing insights into the role played by miRNAs in phenotype development, as well as evolution of gene regulatory circuits in cichlids and other vertebrates [21]. Moreover, these data have already allowed us to study the plasticity of the miRNA transcriptome which produces variable sequences, owing to arm-switching and isomiR generation events, as well as sex dimorphic phenotypes between Nile tilapia genders.

The miRTil database presents 368 precursors and 523 mature miRNAs (high confidence) for Nile tilapia. If we compare these numbers with those for other better-annotated species, such as zebrafish and *Salmon salar*, miRBase offers 346 and 371 precursors and 498 and 350 mature miRNAs, respectively [35]. When we observed specific Cichlidae miRNA detection, Brawand et al. detected 259–286 miRNAs per Cichlidae (*O. niloticus* included) [21], consistent with annotation included in the miRTil, which gathered 271 known precursors. However, putative novel miRNAs in miRTil increases this number to up to 368 miRNAs loci, a significant increase over Brawand et al., who discovered 40 putative novel miRNAs [21]. This data variation was explained as differences in the experimental design [24]. These results indicate that the miRNA identification method (for more details, see Pinhal and Bovolenta et al. [24]) was consistent with the other studies and that the number of Nile tilapia miRNAs annotated is within those expected for Teleostei species. In addition, the miRTil database expands the repertoire of novel miRNAs.

We can also compare miRTil to other plant species databases, e.g., FmMiRNADb, AppleGFDB, and WMP. FmMiRNADb offers information about 355 mature miRNAs for Foxtail millet and molecular characteristics, sequences, annotation, physical genomic position (maps of genomic coordinates), and secondary structure of miRNAs. FmMiRNADb also offers a comparative genome map among the physically mapped miRNAs of foxtail millet and sorghum, maize, rice, and Brachypodium [25]. Similar analyses were performed by our group using other species of Cichlidae, and, although not accessible through miRTil, the results can be found in Pinhal and Bovolenta et al. [24].

The AppleGFDB provides information on genomic location, target mRNAs, pre-miRNA, and mature miRNA sequences for 165 apple mature miRNAs. This database also offers information about transcription factors, proteins, and Gene Ontology (GO) annotation and structure visualization of the miRNA precursor [26].

The WMP, another plant reference database, focuses on showing miRNA expression level differences among environmental conditions (biotic and abiotic stress), including pre-miRNAs sequence, target mRNAs, and expressed sequence tags (ESTs), as well as GO annotations of the target mRNAs.

An interesting aspect of WMP is the impressive number of annotated miRNAs, around 5036 molecules. Considering the evolutionary differences among Metazoa and Viridiplantae, the higher number of miRNAs can be partially explained due to a different methodology to classify miRNAs variations. In WMP, miRNAs reads with nucleotide variations were considered different miRNAs. By applying the miRTil methodology, most of these reads would be classified as isomiRs, a term created to refer to those sequences that have variations concerning the reference mature miRNA sequence [40], e.g., miRNAs reads with few nucleotide variations possibly originated from the same miRNA loci.

However, there are some aspects in WMP and AppleGFDB that inspire future updates in miRTil, such as the presence of a rich graphical visualization of integrated data (WMP) and the GO annotation of the target genes (both), features still absent in miRTil. It is important to observe that the data included in miRTil follows more strict and standardized annotation steps, using state-of-the-art bioinformatics approaches.

## 4. Materials and Methods

### 4.1. Database Structure and User Interface

We built the database structure using the Entity-Relationship Model (ER) following the guidelines for relational database modeling (requirement analysis, conceptual database design, database management system selection, implementation, and testing phases) [41]. The miRTil database structure was implemented in a Database Management System PostgreSQL version 9.0 using SQL language scripts that were generated at CaseStudio v.2 (currently known as Toad™ Data Modeler, Quest Software, Aliso Viejo, CA, USA).

The web interface was constructed using Java web development platform. This platform contains an Integrated Development Environment (IDE) (Netbeans v. 7.3.1, Oracle Corporation, Redwood City, CA, USA), the Java Development Kit (JDK), an application server (Apache Tomcat v. 7.0.42, Wakefield, MA, USA), and an interface-to-database communication framework based on object-oriented programming (EclipseLink v. 2.4.2, Ottawa, Canada).

### 4.2. Database Content

MiRTil database was populated using data generated by our research group (partially presented in Pinhal and Bovolenta et al. [24]). The database integrates results from multiple experimental molecular and in silico analyses, including miRNA identification and expression, target prediction, and genomic localization.

The data related to miRNA identification and expression were derived from the analysis of miRNA sequences obtained by RNA-seq (RNA-seq, Illumina GAIIx platform) and RT-qPCR. RNA samples were obtained from the following tissues: female and male brains, gonads and red and white skeletal muscle, female heart, and eyes and liver tissues from mixed-gender from adult individuals (6-month-old fish). We also included data from different developmental stages (2 days post-fertilization [dpf], 3 dpf, 4 dpf, 5 dpf, and 10 dpf) (GEO accession: GSE102878). Details about the experimental workflow, pre-processed data stages, miRNA characterization, annotation, and quantification can be found in Pinhal and Bovolenta et al. [24].

### 4.3. Target Prediction

To perform the target prediction, we used the TargetScan v.6 [37], Nile tilapia miRNAs, and 3’ UTRs from Ensembl v.79 [36]. We only considered 3’ UTRs from orthologous genes among Nile tilapia, zebrafish, and humans.

However, as there is no complete Nile tilapia 3’ UTR annotation among the orthologous genes, we developed an in-house strategy to expand this repertoire based on stop codon annotations. For genes with an annotated stop codon, we considered, as 3’ UTR, the sequence extracted from the 500 nucleotides downstream of the last stop codon. For genes lacking an annotated stop codon, we considered as 3’ UTR the 500 nucleotides upstream of the last nucleotide of the coding DNA sequences, including, partially, the last exon. The stop codon annotations and coding DNA sequences were obtained from Ensembl [36]. The 3’ UTR sequence length (500 nucleotides) was estimated based on previous studies evaluating the specificity and sensitivity of target prediction using different 3’ UTR lengths in zebrafish (to be published elsewhere) and on 3’ UTR length without overlap among annotations.

In the prediction step, we submitted the 3’ UTR sequences of each species and the mature miRNAs of Nile Tilapia to the prediction algorithm separately. In this step, our approach considered the conservation among miRNA-mRNA interactions, i.e., an mRNA was considered as a miRNA target if the corresponding miRNA-mRNA interaction could be detected in all three species.

Finally, we included another step to guarantee the high confidence of the putative miRNA-mRNA interactions based on the ranking of TargetScan context+ score, an internal TargetScan score based on features of miRNAs seeds and paring between miRNAs-mRNAs that rank the probability/force of the interaction [29].

miRTil also presents tissue and developmental expression profiles for target genes. This data was obtained using data from a RNASeq evolutionary study of African cichlids [21]. We collected the publicly available raw data from the NCBI short read archive (SRA) (accession number SRP009911) and then estimated the gene expression level using the RSEM software on each correspondent sample [42]. Transcripts with fragments per kilobase of transcript per million mapped reads (FPKM) larger than five were considered as expressed and tissues or developmental stages descriptions were included in the miRNA-mRNA interaction annotation.

### 4.4. Genomic Structure and Context

To create the miRNA genomic coordinate map, we aligned the predicted pre-miRNA sequences to the Nile tilapia genome (oreNil2) using PaTMan [43]. Furthermore, we incorporated data obtained from Huang et al. and Yan et al. that were submitted to mapping using miRDeep2 to identify genomic coordinates of the pre-miRNA genes [18,22,44]. All coordinates were annotated and highlighted on the miRNAs genomic coordinate maps. We also included possible miRNA cluster formation (tandem genes) in this map. Additionally, information about miRNA loci, such as genomic location in either inter- or intragenic regions (exonic or intronic miRNAs), was included with information on their host genes taken from Ensembl [36]. Further information on data annotation steps can be found in Pinhal and Bovolenta et al. [24].

## 5. Conclusions

In conclusion, we developed a comprehensive database for bona fide Nile tilapia miRNAs. The miRTil database is accessible through an open-access user-friendly web interface. In the future, we will incorporate GO terms of miRNA targets and synteny analysis similar to the one offered by Genomicus [45]. This data will allow the representation of miRNA conserved genomic blocks among vertebrate species and a miRNA centered view of the evolutionary forces present on Nile tilapia lineages and other teleosts.

## Figures and Tables

**Figure 1 cells-09-01752-f001:**
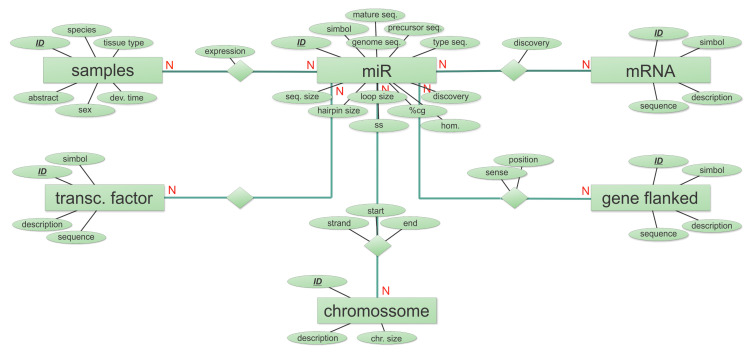
Entity-Relationship Diagram (ERD) for the miRTil database. Squares symbolize entities, a collection of similar data that can be distinguished and have stated relationships to other entity data; ellipses represent attributes, characteristics of stored data. Attributes can be used to guarantee data individuality (primary key); they also establish a relationship with other entities (foreign key). Primary keys are represented by underlined descriptors; diamonds represent relationships between two entities. In addition, rules of cardinality are highlighted in the ERD.

**Figure 2 cells-09-01752-f002:**
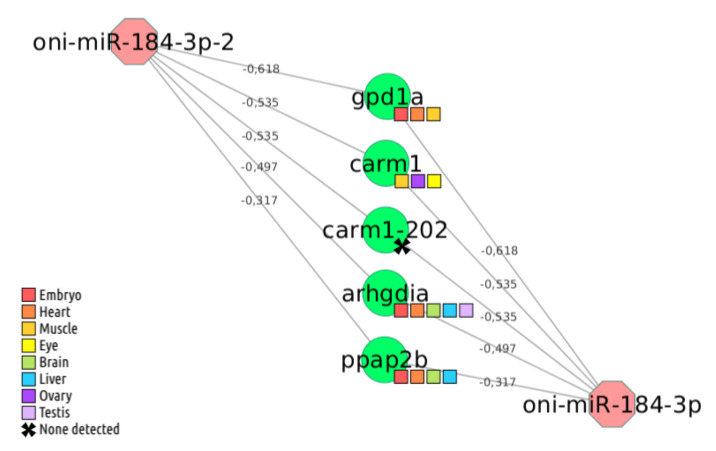
Predicted target network interactions of oni-miR-184-3p and oni-miR-184-3p-2. Red octagons symbolize the miRNAs, green circles the target genes, and gray lines represent predicted interactions by TargetScan v6.0 following the context+ score values over lines (lower is better). Target gene expression was estimated using public transcriptome datasets, and its presence in the sample was represented by colored squares (see Materials and Methods, Target Prediction section).

**Figure 3 cells-09-01752-f003:**
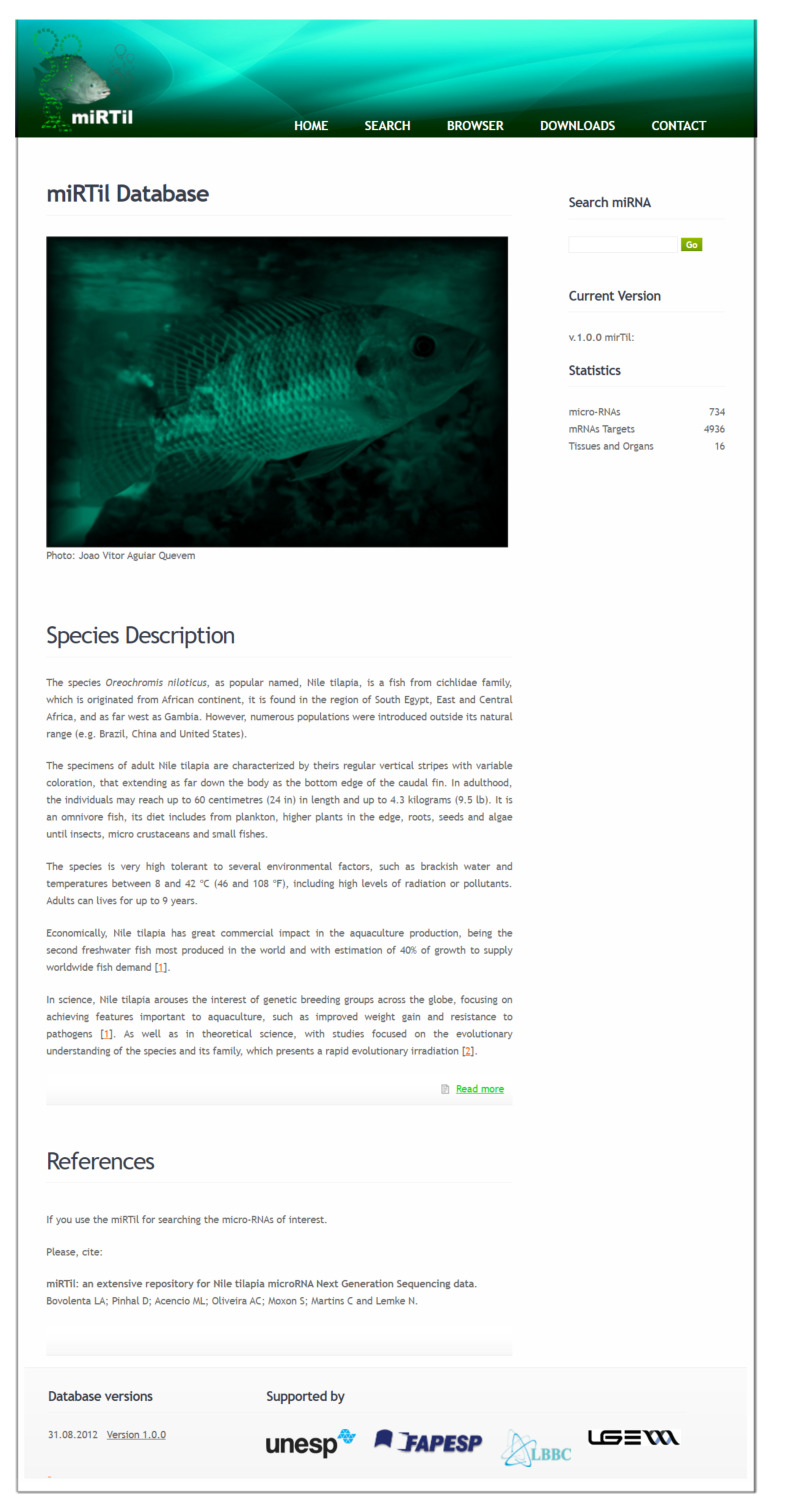
MiRTil database. The home page contains a brief description of Nile tilapia and microRNA (miRNA) data statistics, as well as the ability of a quick search of miRNA data, links to developmental and support groups, old versions of the database, and menu of functionalities.

**Figure 4 cells-09-01752-f004:**
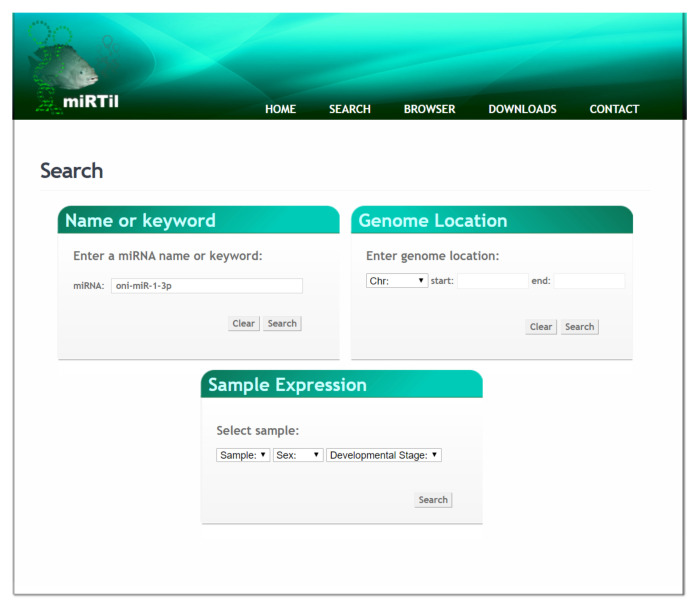
Advanced search mechanism page. The page contains three types of search mechanisms: miRNA symbol name or ID, genomic location, or identified sample.

**Figure 5 cells-09-01752-f005:**
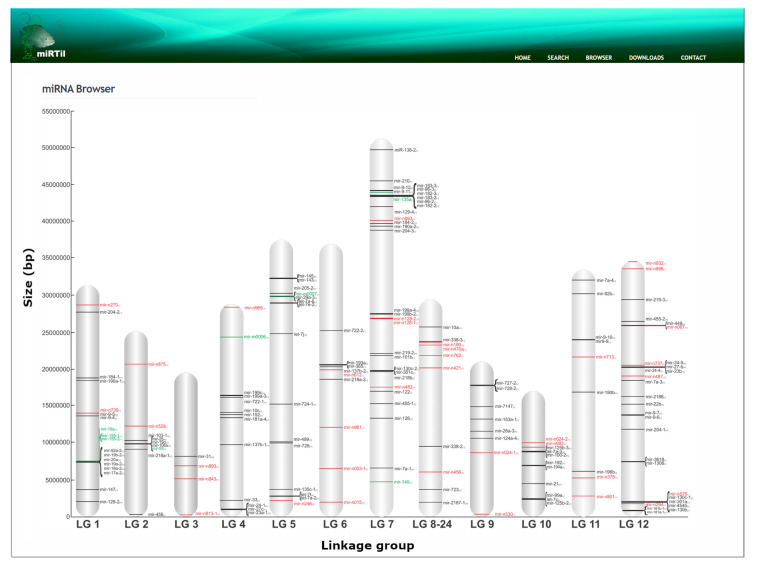
Genomic map of Nile tilapia miRNAs. Red symbols represent novel miRNAs; grey symbols represents conserved miRBase miRNAs [35]; green symbols represent miRNAs identified by other research groups in Nile tilapia. The x-axis shows linkage groups (LGs) from the Nile tilapia assembly (oreNil2), and the y-axis shows the size of LGs (base pair). All miRNAs represented in a single locus or cluster are linkable to detailed miRNA information.

**Figure 6 cells-09-01752-f006:**
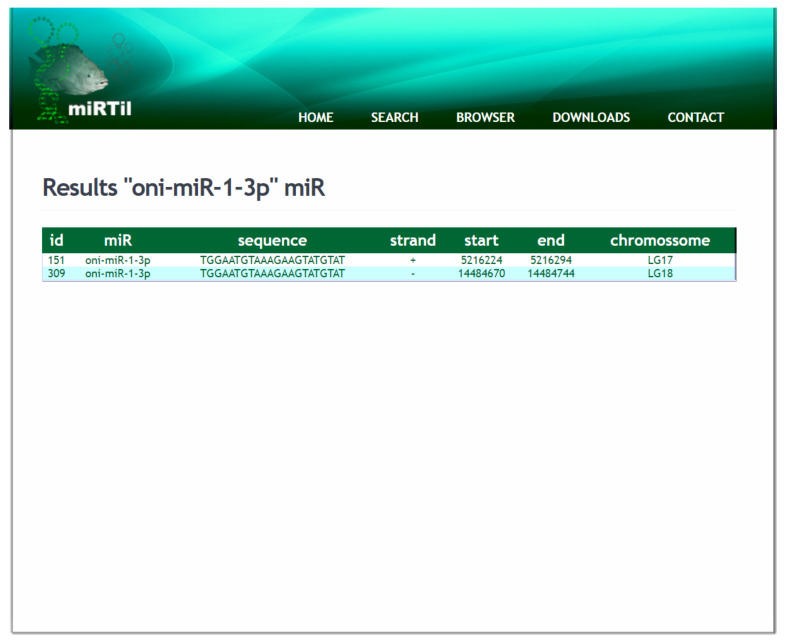
Simple miRNA information result. The page containing summarized miRNA information, such as miRNA ID, symbol, and mature sequence, as well as genomic location details.

**Figure 7 cells-09-01752-f007:**
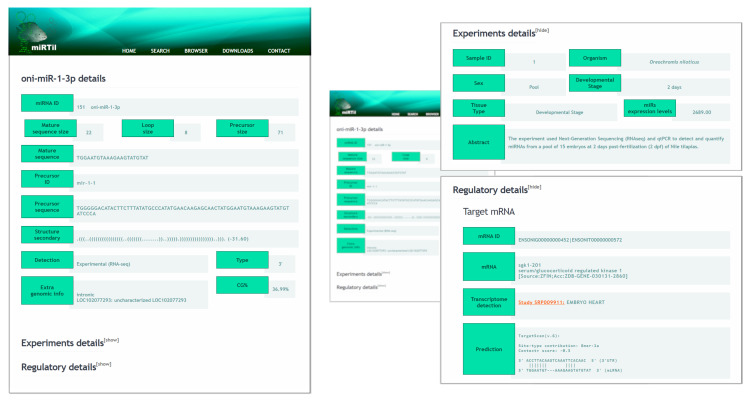
miRTil detailed results pages. The expanded page containing detailed miRNA information, such as identification and classification information, including structural and genomic location information. Extra menus can access hidden details about miRNA experimental and regulatory roles.

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
