# Peer review of "miRTil: An Extensive Repository for Nile Tilapia microRNA Next Generation Sequencing Data"

_cells, 2020, doi:10.3390/cells9081752_

Round 1

Reviewer 1 Report

Thank you for submitting the manuscript titled "miRTil: an extensive repository for Nile tilapia microRNA Next Generation Sequencing data." Clearly an immense amount of work has been done in order to obtain 11 distinct tissues and five key developmental stages of the Nile tilapia and associated miRNAs. The benefit of a database (miRtil) as extensive as this, and freely available, is a tremendous benefit species enhancement.  I went onto the site and found it easy to navigate and use. This manuscript was a pleasure to read.

Line 47. Revise sentence structure.

Line 48. Re structure sentence. Such as e.g. species of origin, molecule identification (precursor and mature sequence details), genomic location etc. Sentence is too long.

Line 51. Missing word between 'allows and obtain'. Additionally the sentence needs revising for soundness.

Line 54. Please spell out the acronym 'MFE'.

Line 54. Please explain what is meant by 'possible information'.

Legend Figure 1. I am unable to see anything red in the figure "In addition, rules of cardinality are highlighted in the ERD".

Line 69. Change ≈ to approximately.

Line 78. Remove the 's' from the word 'interactions'.

Legend Figure 2. Remove the word 'to' in this sentence 'green circles to the target genes'.

Legend Figure 2. Remove the word 'to the' in this sentence "gray lines to the interactions predicted" and replace with 'grey lines represent predicted interactions'.

Legend Figure 2. RE structure this sentence "Expression presence of target gene in transcriptome samples were checked with public the data set."

Legend Figure 3. I have reworded this sentence here. "The home page contains a brief description of Nile tilapia and miRNA data statistics as well as the ability of a quick search of miRNA data, links to developmental and support groups, old versions of the database and menu of functionalities."

Legend Figure 4. Please revise in line with the one above.

Legend Figure 5. Please add in axis titles to the image.

Line 148. Should 'use' be 'user'?

Line 149. Disjointed sentence.

Line 156. Replace the word 'Once'.

Line 159. Remove 's' on miRNAs.

Line 159 This is not a sentence "In the advanced content page (Figure 7)."

Line 165. Add the word 'number' after accession.

Line 185. Remove the 's' in miRNAs.

Line 204. Revise the sentence beginning with "These results..." as the message in the sentence is not clear.

Line 221. This sentence needs revising. "Here, we highlighted for uses of annotation methodology applied in miRTil data."

Line 228. Remove the 's' in futures.

Line 245. open ended bracket.

Line 253. Remove the 'es' in fishes.

Line 253. Remove the 's' in months

Line 282. Please spell out the acronym "FPKM".

Line 289. Remove the 's' in clusters.

Reviewer 2 Report

The authors presented an open repository dedicated to Nile Tilapia miRNAs named the miRTil database. The database stores data on 734 mature miRNAs identified in 11 distinct tissues and five key developmental stages, providing detailed information about miRNA structure, genomic context, predicted targets, expression profiles, and relative 5p/3p arm usage. This database also includes a comprehensive pre-computed miRNA-target interaction network containing 4,936 targets and 19,580 interactions. This research is interesting and novelty, it will contribute to facilitate the implementation of automated workflows and other large-scale analysis necessary to provide Nile tilapia-specific miRNA data in an open-access and user-friendly database. Therefore, I think this paper can be accepted to publish in its present form.
